# A Lake Extraction Method Combining the Object-Oriented Method with Boundary Recognition

Bingxue Liu [1,2], Wei Wang [1,*] and Wenping Li [1,2]

1 State Key Laboratory of Resources and Environmental Information System, Institute of Geographic Sciences and Natural Resources Research, Chinese Academy of Sciences, Beijing 100101, China
2 University of Chinese Academy of Sciences, Beijing 100049, China
* Correspondence: wang_wei@lreis.ac.cn

**Abstract:** The China–Pakistan Economic Corridor is the pilot area of the Belt and Road, where glaciers and lakes are widely distributed. Recent years, global warming has accelerated the expansion of glacier lakes, which increased the risk of natural disasters such as glacier lake outburst. It is important to monitor the glacier lakes in this region. In this paper, we propose a method combining the object-oriented image analysis with boundary recognition (OOBR) to extract lakes in several study areas of China–Pakistan Economic Corridor (CPEC). This method recognized the lake boundary with the symmetrical characteristic according to the principle of seed growth of watershed algorithm, which can correct the boundary extracted by the object-oriented method. The overall accuracy of the proposed method is up to 98.5% with Landsat series images. The experiments also show that the overall accuracy of our method is always higher than that of the object-oriented method with different segmentation scales mentioned in this paper. The proposed method improved the overall accuracy on the basis of the results obtained by the object-oriented method, and the results with the proposed method are more robust to the seeds than that with the boundary correction method of the watershed algorithm. Therefore, the proposed method can obtain a high extraction accuracy while reducing the complexity of the object-oriented extraction.

**Keywords:** landsat; object-oriented; symmetrical boundary recognition; seed growth; watershed algorithm; lake extraction

## 1. Introduction

In the context of global warming, the glacier melting has accelerated the glacier lakes expansion, which may cause many related geological disasters. In the area with human activities, that will pose a threat to the safety of human life and property. The China–Pakistan Economic Corridor region is the pilot area of the Belt and Road. There are many glaciers and lakes in this region. In the upstream region of the China–Pakistan Economic Corridor, various glacier related disasters, including glacier lake outburst, are particularly prominent among numerous mountain disasters, posing a threat to the China–Pakistan Highway, the infrastructure and human activities [1–5]. Therefore, it is significant to monitor the glacier lakes in real time in this region. With the continuous development of remote sensing technology, the method of using remote sensing data to extract surface features has gradually replaced the traditional method of artificial survey.

Recent years, with the rapid development of artificial intelligence, the technology of deep learning has been widely applied in the field of ground feature extraction, these studies have achieved high accuracy [6–12]. Deep learning technology is a "black box" method, which is based on the induction of samples and prediction according to the obtained rules. This process has low interpretability, because of which, the quality of the results of deep learning methods is usually proportional to the sample size and the model trained based on a specific experimental range has low generality. Compared with other

non-"black box" methods, taking the object-oriented method as an example, it classifies and extracts the target ground objects according to certain features, which have some representative meanings that can be understood by human beings, such as water body index. The values of these features are usually stable within a certain range, making the method of classification based on these features more universal, and there is no strict requirement on the training sample size.

The object-oriented extraction method has gradually replaced the pixel-level extraction method in the lake extraction with remote sensing data [13]. The object-oriented image analysis (OOIA) is based on the "bottom-up" region merging principle. In addition to the spectral features of the image, it also pays attention to the shape factors of the ground objects. In the existing research, the application of object-oriented methods to the extraction and classification of ground objects has been relatively successful [14–17], one of which is the lake extraction in the Kanas Lake area of Altay Mountain, Xinjiang, the accuracy of which is more than 95% under cloudless conditions using the object-oriented method [18]. However, the classification accuracy of the object-oriented method can be easily affected by the segmentation scale and the feature rule set, and the misclassification often occurs at the boundary of different ground objects. Improving the classification accuracy of pixels near the lake boundary can effectively improve the overall accuracy of lake extraction.

According to the intensity of the change of gray values, the boundaries in the image are usually divided into two categories: "step" shape and "roof" shape. For a point on the boundary with the "step" shape, the first derivative or the gradient of its gray value change curve reaches the maximum at this point [19]. Traditional gradient-based boundary detection operators include Roberts, Sobel, Prewitt, etc. The application of these methods in image edge detection has achieved good results [20–24]. However, the boundary response in the traditional edge detection algorithm based on the first derivative is wide, which makes the boundary location inaccurate [25–28]. The canny algorithm conducts the method of non-maxima suppression to refine the boundary in order to detect the boundary more accurately, which is also widely used in the field of image edge detection [29–31]. In the study of boundary enhancement and refinement, a symmetry attribute was also proposed to enhance texture, boundary and micro structure in seismic profiles [32]. Using the symmetry attribute, the extremum of seismogram can be effectively sharpened, which makes the boundary information in the profile more clearer.

It is efficient to detect all the boundaries in one image by using the mentioned edge detecting algorithms. However, for the edge extraction of one certain kind of object in the image, it is necessary to determine the location of the object previously, after which the edge detection algorithm can be used to improve the accuracy of the object extraction. Watershed algorithm is an image segmentation algorithm based on the principle of seed growth [33], which has been widely used in object edge detection in recent years [34–37]. The overall accuracy of lake extraction in [38] has reached more than 90% with the method of combining object-oriented with watershed algorithm. However, for the problem of extracting the boundary of one kind of object, the extraction result of watershed algorithm is highly dependent on the seed; thus, the result using watershed algorithm is not stable enough.

The pixel gray value change curve at the lake boundary is a typical "step" shape and the pixel values in the neighborhood of the lake boundary appear with the characteristic of symmetry in its first derivative image. It is worthwhile to explore the application of symmetrical boundary enhancement method in the extraction of ground objects, so we propose a lake extraction method based on combination of object-oriented image analysis and boundary recognition (OOBR). We firstly extracted the lake with the object-oriented method, then modified the lake boundary by combining the seed growth principle of watershed algorithm with the symmetrical boundary recognition.

The main content of this paper includes the selection of parameters and the experiments for verifying the proposed method. The experiments are designed to verify the method of symmetrical boundary recognition based on the seed growth can really increase the extraction accuracy from the results from the object-oriented method. In addition, we

analyzed the impact of different segmentation scales on the extraction accuracy of our method and the object-oriented method; we also compared the results obtained using the watershed algorithm with those obtained using the method of symmetrical boundary recognition. The analysis in this paper will verify the high overall accuracy of the proposed method and its robustness to the seeds in the lake extraction.

## 2. Materials and Methods

### 2.1. Materials

#### 2.1.1. Study Region

Research statistics show that the upstream area of the China–Pakistan Economic Corridor (CPEC) is particularly prominent in various geological disasters related to the evolution and dynamics of glaciers [5]. We selected the upstream area of the China–Pakistan Economic Corridor (CPEC) as the study region, where mountains are developed, glacier lakes are numerous and concentrated in those regions. In this paper, we selected three regions where glacier lakes are developed for experiments. Figure 1 shows the general geographic situation of the study regions.

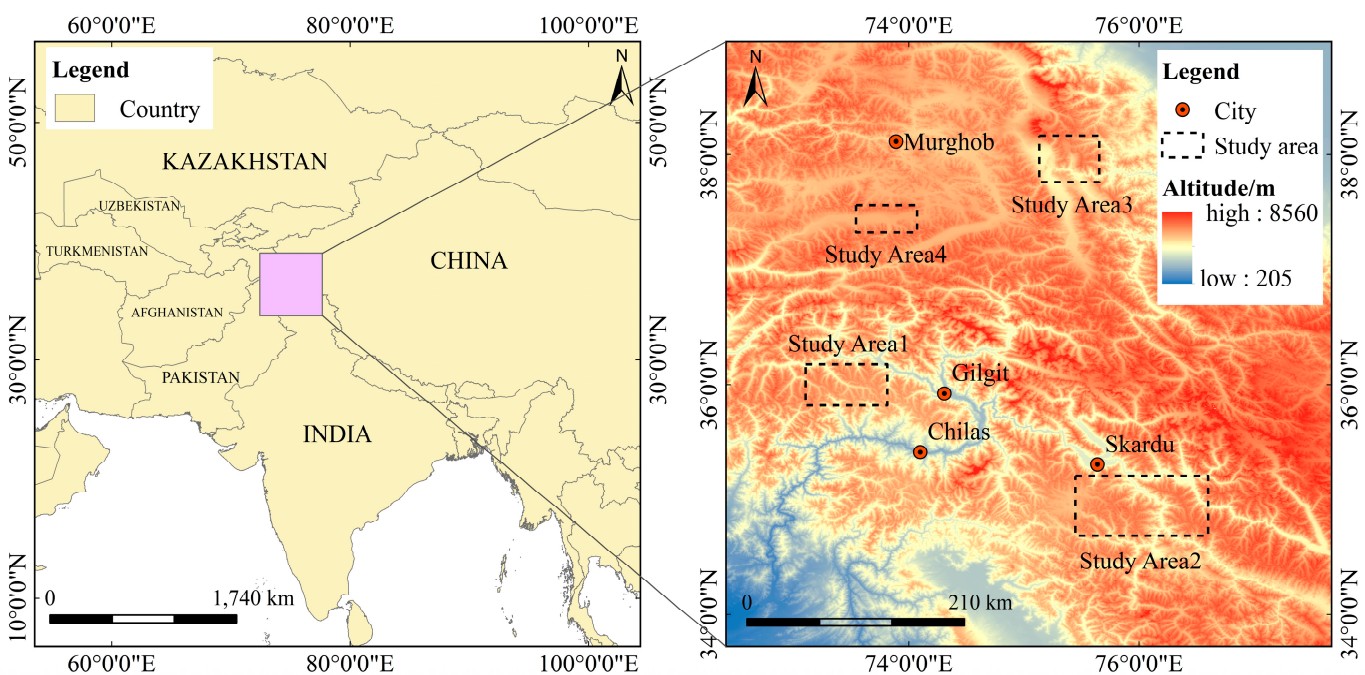

**Figure 1.** General geographic situation of the whole study area.

The longitude of the whole study area ranges from 73° E to 76.6° E, and the latitude ranges from 34.6° N to 38° N. The regional elevation ranges from 651 m to 8560 m, and the average elevation is 4605.5 m. The whole study region spans five countries: China, Pakistan, Tajikistan, Afghanistan and India. The annual average temperature in this region is about 9 °C, while the climate is cold and dry. The study area 1 and 2 are located outside the boundary of China, in the upstream of the China Pakistan Economic Corridor; the study area 3 is in the south of Kashgar, Xinjiang, China; study area 4 is used for the verification of the universality of proposed method.

#### 2.1.2. Preparation and Preprocessing of Data

The required images are from the Geospatial Data Cloud Platform (https://www.gscloud.cn/, accessed on 22 October 2022) [39] and the USGS (https://earthexplorer.usgs.gov/, accessed on 22 October 2022) [40]. The data include STRM 1 DEM (the resolution is 30 m) and Landsat series images: Landsat TM/ETM+/OLI. Table 1 shows the information

of the Landsat series images selected. We used these images to determine the parameters, verify the effectiveness of the method and carry out comparative experiments between different methods.

**Table 1.** The information of remote sensing images.

| Image | Path/Row | Date | Cloud Cover/% | Sensor |
|---|---|---|---|---|
| LE7_4 | 150/34 | 15 August 1999 | 3.00 | ETM+ |
| LE7_1 | 150/35 | 16 September 1999 | 1.00 | ETM+ |
| LT5_2 | 148/36 | 27 August 2000 | 2.00 | TM |
| LE7_3 | 149/34 | 30 September 2001 | 1.00 | ETM+ |
| LT5_1 | 150/35 | 2 October 2008 | 1.00 | TM |
| LT5_3 | 149/34 | 27 August 2009 | 2.00 | TM |
| LT5_4 | 150/34 | 24 August 2011 | 1.00 | TM |
| LC8_1 | 150/35 | 19 August 2015 | 1.42 | OLI |
| LC8_2 | 148/36 | 21 October 2020 | 0.71 | OLI |
| LC8_4 | 150/34 | 4 September 2021 | 1.83 | OLI |

In order to effectively verify the performance of the proposed method, the cloud cover of the selected Landsat series images is less than 3%, and the date of image requirement ranges from August to October; this is because the average temperature of the study area during which is higher throughout the year, the frozen lakes are fewer, and the rate of snow coverage is lower, which can effectively reduce the interference of snow.

Three types of Landsat images of study area 1 and study area 4 were obtained. Since before the failure of the scan lines corrector (SLC) of Landsat7 ETM+, there was no remote sensing image of study area 2 meets the requirement that the cloud cover is less than 3%, only the Landsat TM and Landsat OLI images of study area 2 were obtained. The images of study area 3 used for the parameter selection include Landsat TM and ETM+.

*2.2. Methods*

We proposed a method (OOIA + Boundary Recognition, OOBR) to extract the lakes in the China–Pakistan Economic Corridor (CPEC) region in this paper, which is divided into two steps: (1) selection of seeds: preliminary extraction of lakes by the object-oriented method; (2) correction of boundary: the correction of the lake boundary by symmetrical boundary recognition. The method can be also seen as: feature extraction using object-oriented method and post-classification process based on boundary recognition. Figure 2 shows the technology route of the proposed method (OOBR).

2.2.1. Feature Extraction: Selection of Seeds

The process of feature extraction of lakes by the object-oriented method includes two steps: multi-resolution segmentation and assign-class. The object-oriented image analysis incorporates spectral, textural, contextual and pattern information in the classification process [41].

The multi-resolution segmentation is based on the principle of "maximum heterogeneity". Image pixels are continuously aggregated until the heterogeneity of the aggregated "object patch" is greater than a specific threshold. In this process, the original image can be segmented into multiple "object patches", after which its basic unit changes from pixels to "object patches", the spectral characteristics of pixels within the same "object patch" are similar. In the experiment, the slope and mountain shadow of the study area are also taken as the spectral factors, the weights of which are both set to 0.5, and the weight of the near-infrared band is set to 2. The weights of the other bands, the shape factor and the compactness remain as their default settings.

Based on multi-resolution segmentation, the process of assign-class is selecting appropriate features and thresholds to build a feature rule set and merging the "object patches" to obtain the lake area (seed) and candidate lake area. The requirement of features is that there is a great distinction of the feature values between water and non-water. In this paper, the near-infrared band, slope and appropriate water index are all selected as the features in the feature rule set.

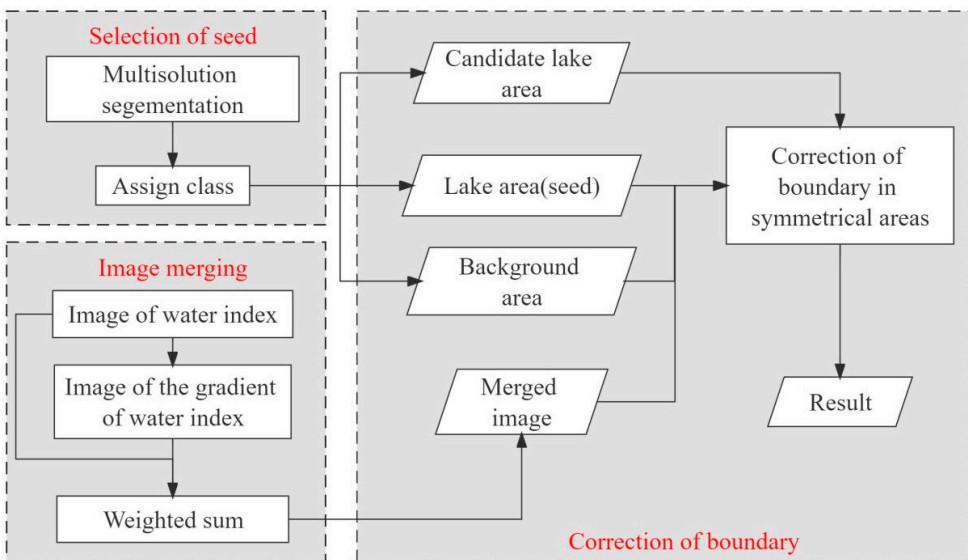

**Figure 2.** Technology route.

2.2.2. Post Classification: Correction of Boundary

The boundary correction method proposed in this paper originates from the symmetry enhancement of boundary: the characteristic curve representing the symmetry of the original seismogram is generated by calculating the characteristic value of symmetry on each point of the seismogram [32]. By combining the characteristic curve with the original seismogram, the points with obvious symmetrical features on the original seismogram can be enhanced. Corresponding to the image of seismic profile, the boundary of stratum and microstructures are clearer, which suggests that the use of symmetry attributes can eventually improve the quality of profile data.

In order to further improve the accuracy of classification near the lake boundary, we propose a boundary recognition method combining the symmetry of boundary and the "seed growth" principle from the watershed algorithm. Based on the preliminary extraction results of lakes by the object-oriented method, we mark different areas separately: background (0), lake areas (1, 2, . . . , n), candidate lake area (−3). After that, we remark the outermost pixels of the lake area as the candidate lake area (−3), the purpose of which is to make the seed regions not cover the symmetrical boundary region. Figure 3 shows the marking process described above.

We carry out the seed growth process in the boundary-enhanced image: searching the eligible pixels in the symmetrical areas as the lake boundary pixels. The boundary-enhanced image is the weighted combination of the water index image and its gradient image, which is inspired by the profile image enhancement principle [32]. In the enhanced image, there are prominent symmetrical areas near the lake boundaries as Figure 4. For different types of seed regions, the process of boundary recognition is different.

We select two thresholds (threshold_1 = 150 and threshold_2 = 145, threshold ∈ [0, 255]) for lake boundary recognition given the weight settings in this paper. It means that if the internal pixel value is greater than 150 and the external pixel value is lower than 145, the external pixel will be marked as the lake boundary. Figure 4 shows four types of seed boundary: a, b, c and d. For type a, the boundary of the seed region is not in the symmetrical area, but it will enter the symmetrical area with the growth of seed, before

which the pixel value may increase or decrease with the seed growth; however, the value is relatively lower than that in the symmetrical area. For type b, the boundary of seed region is located in the rising stage of the symmetrical area, which indicates that it is necessary to continue the seed growth to search the lake boundary. For type c, the boundary of seed region is located outside of the area meeting the recognition thresholds, and the pixel value will decrease with the seed growth, which may also happen in type a; this is why we must let the process of seed growth continue. For type d, the boundary of seed region is outside of the whole symmetrical area. Where the pixel value is zero, it is impossible to find the pixel which meets the recognition thresholds with the seed growth in this case; thus, the seed growth is interrupted directly and the boundary of the seed region is recognized as the lake boundary. In addition to the case of type d, when the boundary of seed is located in the area originally marked as the background, seed growth is also directly interrupted.

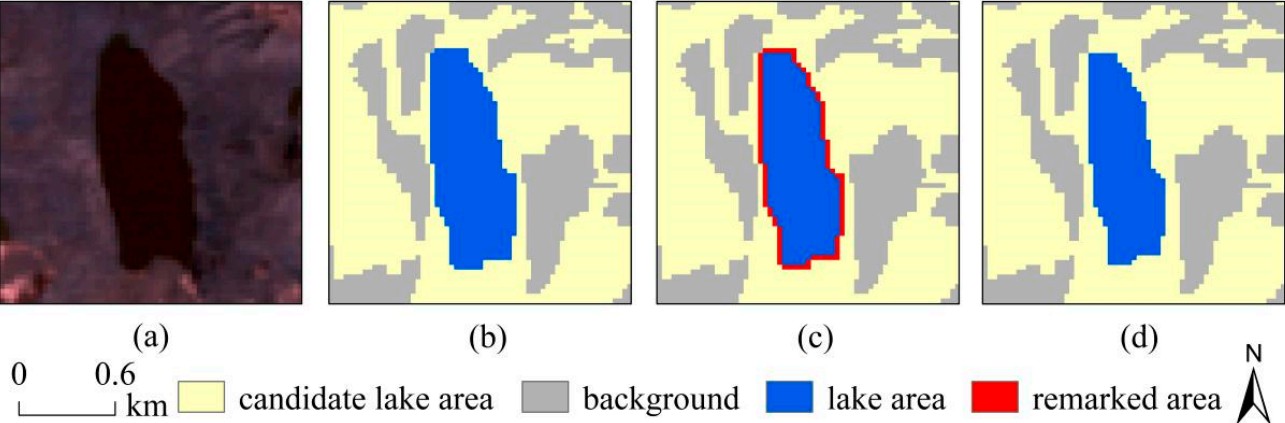

**Figure 3.** Marking process of proposed method: (**a**) Landsat TM image; (**b**) marked result by object-oriented method; (**c**) remarked area (the red pixels) using the proposed method; (**d**) final marked result.

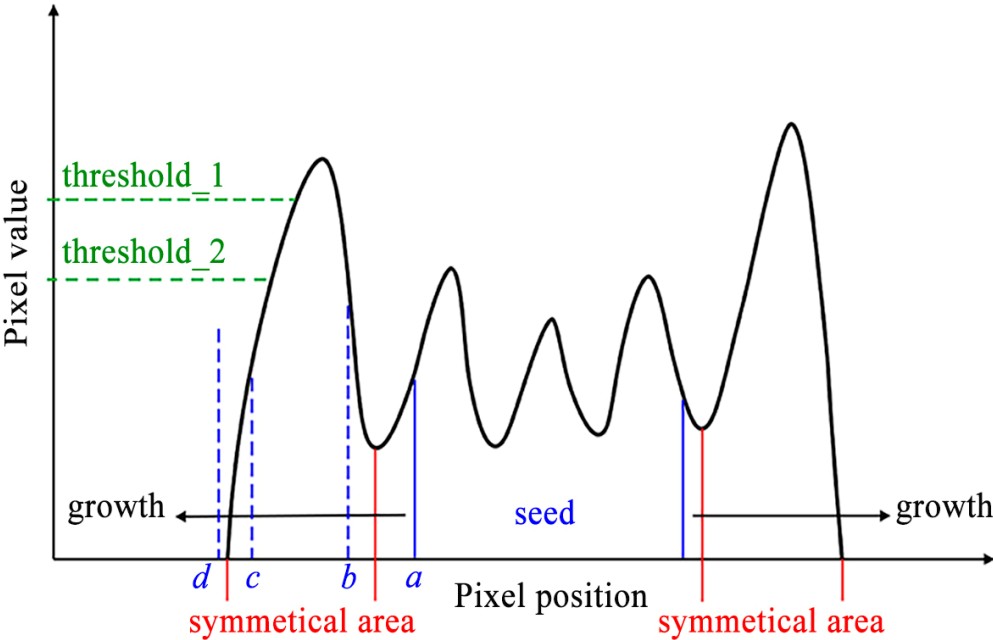

**Figure 4.** Four types of seed regions in the seed growth: a–d represents four typical positions of seed boundary.

*2.3. Setting of Parameters*

2.3.1. Water Index in Feature Rule Set

In order to ensure the simplicity and effectiveness of the feature rule set constructed in assign-class stage, it is essential to select an appropriate water index as a parameter in the rule set. The commonly used water indexes are the normalized difference water index (NDWI) [42] and the modified normalized difference water index (MNDWI) [43]. The definition of the two water indexes is shown, respectively as:

$$NDWI = \frac{green - nir}{green + nir} \tag{1}$$

$$MNDWI = \frac{green - mir}{green + mir} \tag{2}$$

where *green* represents the green band of the image, *nir* represents the near-infrared band and *mir* represents the mid-infrared band. Both of the water indexes can highlight the water area in the image. In order to obtain a more accurate classification result of the feature rule set, we design a comparative experiment to select a more appropriate water index as the feature parameter. We randomly collect samples of different kinds of "object patches", record the value of two water indexes of these samples and draw line graphs according to the collected data. In this experiment, the sample size is 10 and the surface features include water, snow, mountain shadows, bare land and vegetation. The sampling regions are selected from Landsat TM, OLI and ETM+ from study area 1, which correspond to the segmentation scales of 10, 10 and 75, respectively. Figure 5 shows the line graphs.

In the results, the value ranges of NDWI between water and other surface features do not coincide with each other, whereas the value range of MNDWI between water and other surface features coincide with each other in all of the three images. In the result of NDWI, the value range of water is higher than 0.3, whereas those of other surface features are not higher than 0.2 in the Landsat TM image. In the result of NDWI of the Landsat ETM+ image, the value range of water is higher than 0.3 and that of the other surface features are not higher than 0.3. As for the Landsat OLI image, the NDWI value range of water and the other surface features can be roughly separated by 0.1. In the result of MNDWI: in the image of Landsat TM, the value range of water is higher than that of snow on sample 1–5, whereas the relationship is opposite on sample 6–8. In the image of Landsat ETM+, the value of water on sample 7 is lower than that of hillshade on sample 9; In the image of Landsat OLI, the value of water and hillshade at sample 7, 9 and 10 is almost the same. In order to distinguish water from other surface features effectively, we selected NDWI as one of the feature parameters.

2.3.2. Water Index Image for Image Merging

The symmetry attribute of the boundary is originally in the gradient image. The pixels near the boundary in the original image are highlighted in the gradient image. In this paper, we merge the water index image and its gradient image to sharpen the wide highlighted area in the gradient image.

In our method, the water index image is stretched to a pixel depth of 8 bits in proportion. If the non-water surface features with high value of water index are adjacent to or close to the lake area, the highlighted pixels near the lake boundary may become fuzzy in the merged image because of the interference of non-water surface features. We tend to choose the image of water index in which the average difference of the water index between water and non-water surface features is significant to produce the merged image.

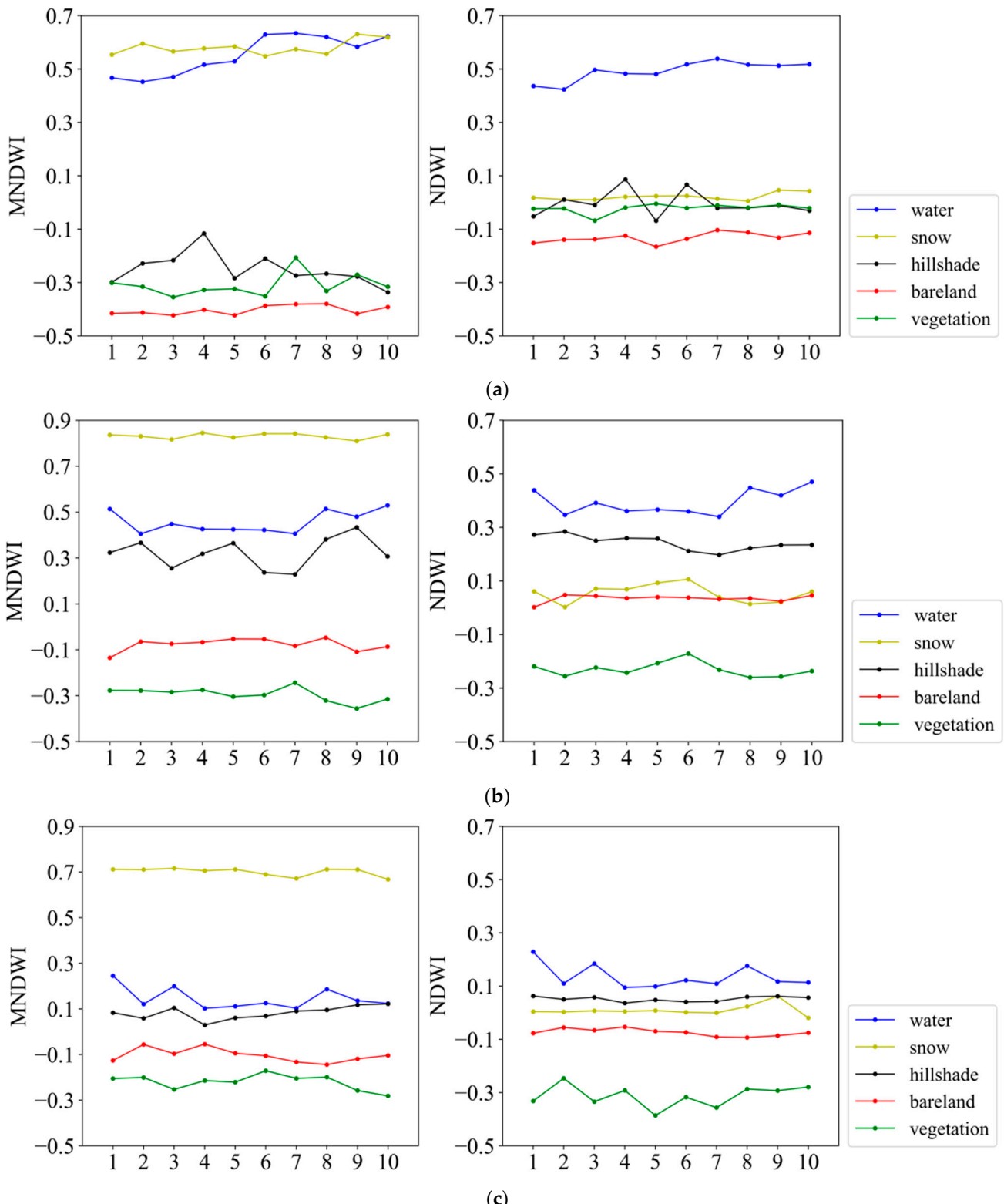

**Figure 5.** Line graphs of water indexes of different surface features: (**a**) MNDWI and NDWI of different surface features of Landsat TM; (**b**) MNDWI and NDWI of different surface features of Landsat ETM+; (**c**) MNDWI and NDWI of different surface features of Landsat OLI.

In this experiment, we calculate the mean values of two water indexes of different surface features using the statistical data in the above experiment and compare the coefficient of variation (CV) of the water index between NDWI image and MNDWI image. Here, we use the calculation method of coefficient of variation defined in terms of statistics:

$$CV = \frac{\sqrt{\frac{1}{n}\sum_{i=1}^{n}\left(\overline{WI_i} - \overline{WI_{water}}\right)^2}}{\overline{WI_{water}}} \tag{3}$$

where $n$ is the number of categories of non-water surface features, and *WI* represents one of the two water indexes. Table 2 shows the calculation results based on the three types of images.

**Table 2.** Coefficient of variation (CV) of water index.

| Image | Coefficient of Variation (CV) | |
| --- | --- | --- |
| | **NDWI** | **MNDWI** |
| Landsat TM | 1.076 | 1.374 |
| Landsat ETM+ | 1.029 | 1.099 |
| Landsat OLI | 1.910 | 2.452 |

According to the results, the CV of MNDWI is greater than NDWI in the three different images. As shown in Figure 6a, it is an area of Landsat TM image selected in the study area 1, and Figure 6b,c are the NDWI and MNDWI images of the same area, respectively. In the NDWI image, the non-water surface objects around the lake form more noise.

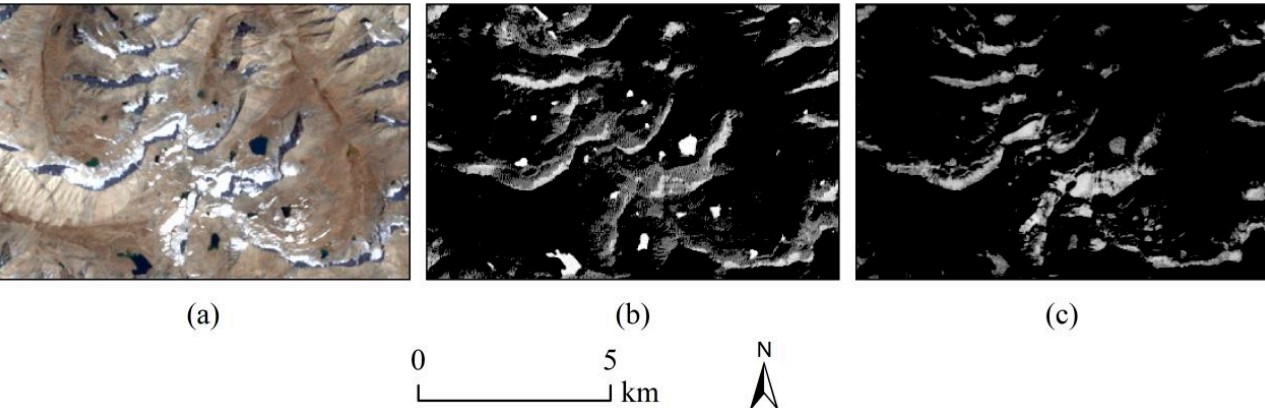

**Figure 6.** Water index image of Landsat TM image: (**a**) Landsat TM image; (**b**) NDWI image; (**c**) MNDWI image.

The coefficient of variation of the two water indexes and the comparison of images suggest that the MNDWI image is more appropriate to produce the merged image, because it can relatively reduce the interference of non-water surface features.

### 2.3.3. Weight of Image Merging

The merging of the image of water index and its gradient image is linear. The appropriate weights of the two images need to be determined before lake extraction. In this experiment, three different lake areas in the three types of images are selected to merge images with different weight settings. The weight settings are, respectively 6:4, 4:6, and 2:8 (water index image: gradient image). Figure 7 shows the merged images.

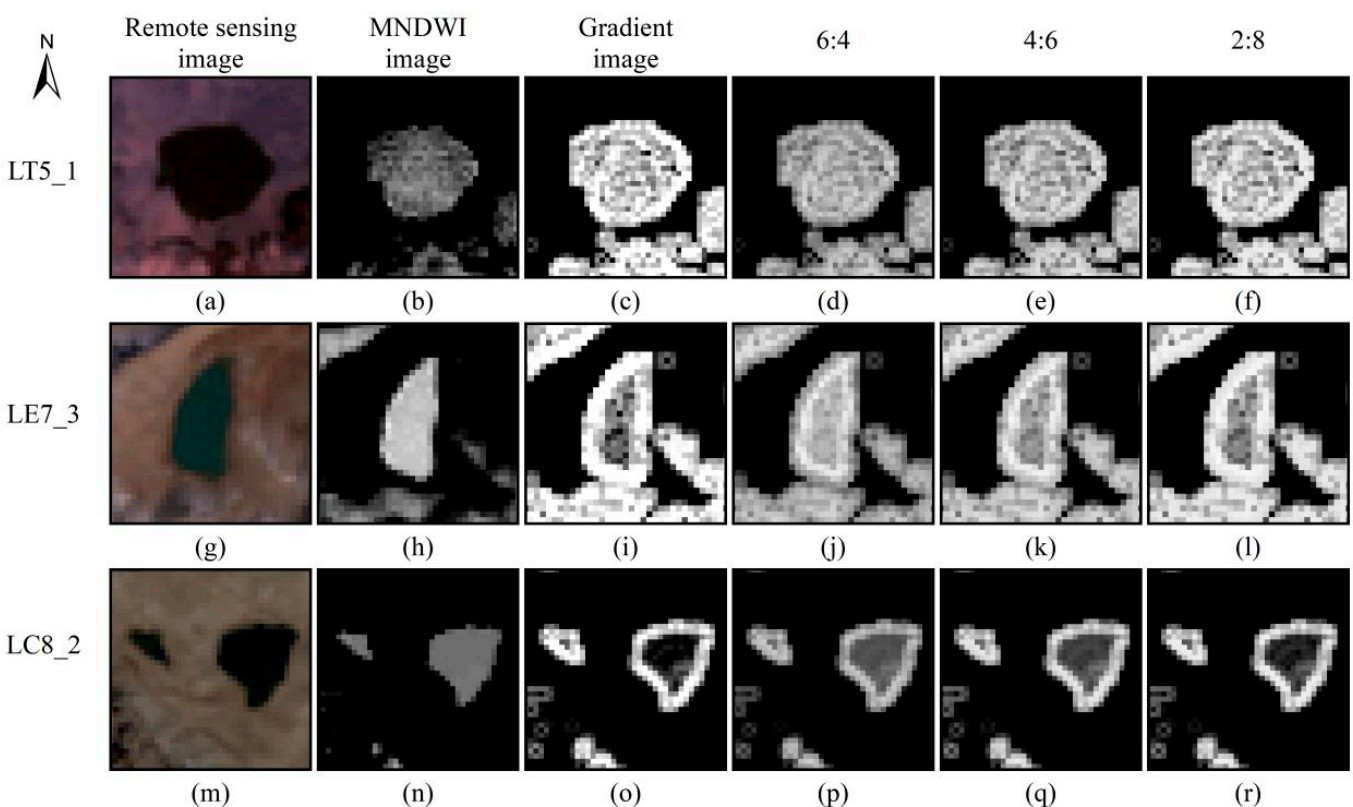

**Figure 7.** Merged image with different merging weights: (**a,g,m**) are from the original remote sensing images of the three experimental areas, respectively; (**b,h,n**) are MNDWI images corresponding to (**a,g,m**), respectively; (**c,i,o**) are gradient images corresponding to (**a,g,m**), respectively; (**d,j,p**) are merged images with the weight setting of 6:4 corresponding to (**a,g,m**), respectively; (**e,k,q**) are merged images with the weight setting of 4:6 corresponding to (**a,g,m**), respectively; (**f,l,r**) are merged images with the weight setting of 2:8 corresponding to (**a,g,m**), respectively.

Comparing the merged images with different weight settings, we find that when the weight is 6:4, the highlight of pixels near the lake boundary in the gradient image is not conspicuous in the merged image. When the weight is 2:8, the highlight in the gradient image is better presented in the merged image, but the wide response of edge in the gradient image is not effectively suppressed, which makes it difficult to recognize the boundary according to the thresholds.

In addition, we select the pixels in one certain row in the images with different weight settings in Figure 7. The "line" composed of this row of pixels "crosses" the lake in the image. In the selected "line", the pixel values near the lake boundary are high. In the process of seed growth from the highest value pixel, the pixel value gradually decreases to 0. We calculate the gradient of decrease in the mentioned descending interval before the pixel value becomes 0 and draw scatter plots with position as the variable as Figure 8.

The response of lake boundary in the gradient image of water index is wide, which means that the change of gradient near the lake boundary is weak. In Figure 8, when the value of blue point is low, the values of red point and green point are both higher than blue point at the same position, which suggests in the merged image, the gradient near the lake boundary where the original gradient is low will increase with the increase in weight of the water index image. Although the values of red point and green point are both lower than blue point at the same position when the value of blue point is high, they are still high overall. The increase in weight of the water index image can effectively suppress the wide response of lake boundary, making it more convenient and easier to set thresholds for lake boundary recognition.

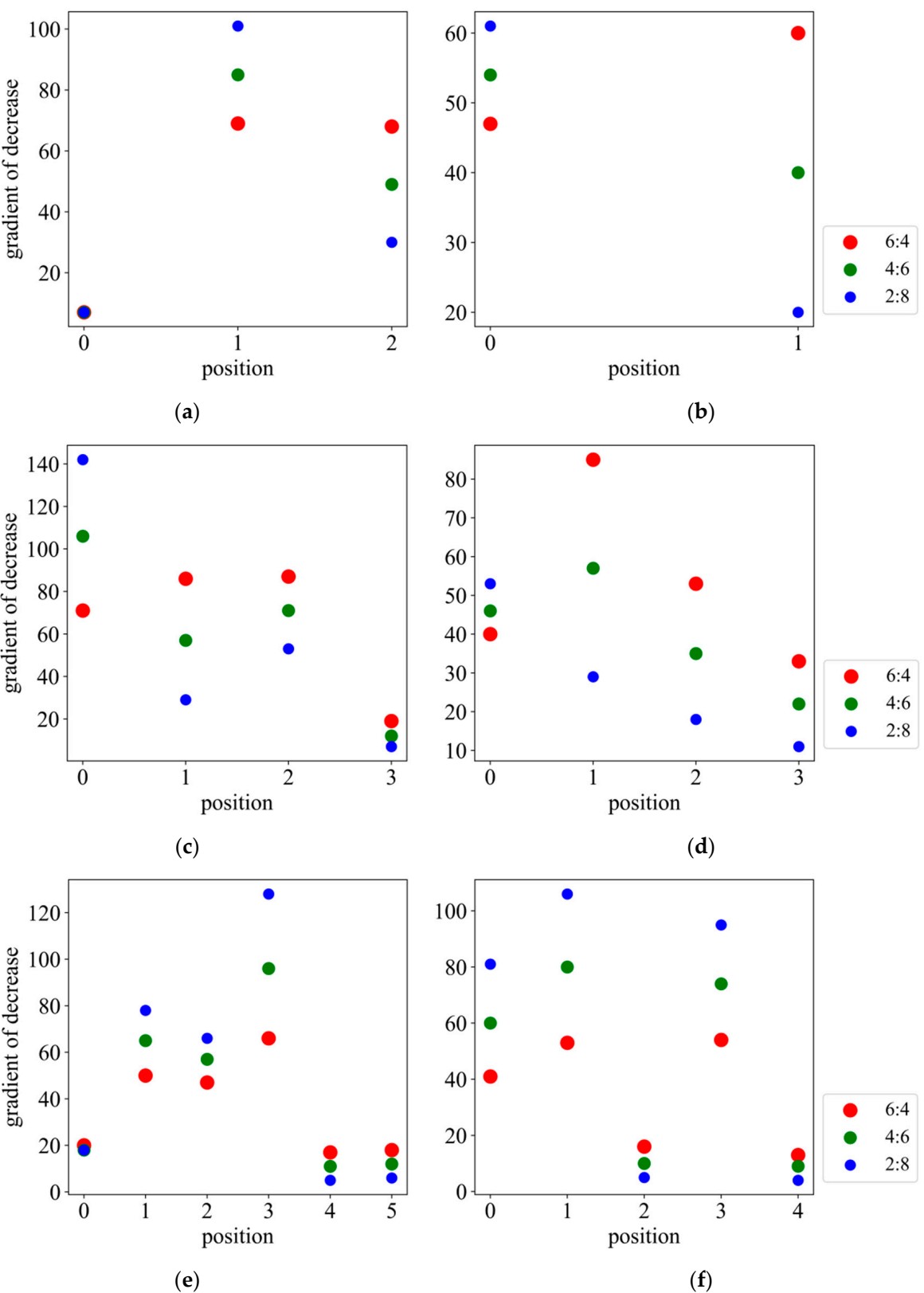

**Figure 8.** Scatter plots of gradient of decrease near the lake boundary with different weight settings: (**a**) result of the 14th row of Landsat TM image; (**b**) result of the 16th row of Landsat TM image; (**c**) result of the 18th row of Landsat ETM+ image; (**d**) result of the 19th row of Landsat ETM+ image; (**e**) result of the 19th row of Landsat OLI image; (**f**) result of the 27th row of Landsat OLI image.

To compare the significance of the highlighted pixels near the lake boundary in the merged images with different weight settings, we calculate the differences between the maximum of the highlighted pixels near the lake boundary and the minimum inside the lake boundary on the selected "lines". Figure 9 shows the results of ranges with different weight settings in the three types of remote sensing images.

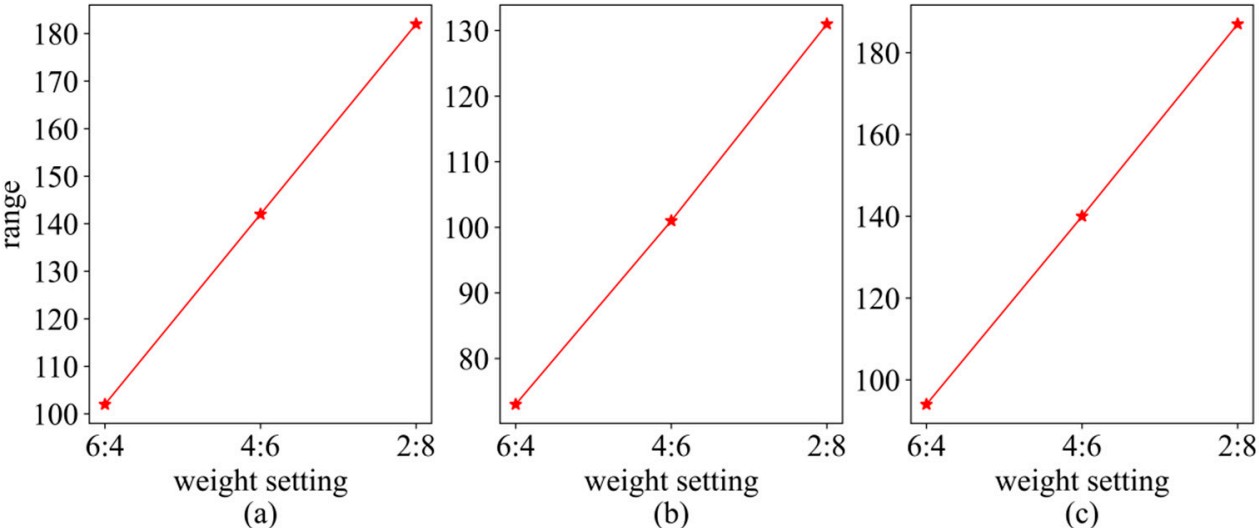

**Figure 9.** Range of values of pixels on the selected "line": (**a**) the range on the 14th row of the image of Landsat TM; (**b**) the range on the 19th row of the image of Landsat ETM+; (**c**) the range on the 27th row of the image of Landsat OLI.

When the weight setting of the image of water index and image of gradient is 6:4, the range of pixel value is the lowest, while when the weight setting is 2:8, the range of pixel value is the highest. The range is decreasing with the increase in the weight of the image of water index. The accuracy of boundary correction relies on the effect of highlight of the pixels near the real lake boundary, if the highlighted pixel values have almost no difference from the pixel values inside the lake, it is difficult to recognize the lake boundary effectively. Therefore, it is better to set the weight of image of water index as not very high.

The two aspects of analyses of the merged images with different weight settings suggest that the weight setting of 4:6 is the best choice for the image merging. It can not only suppress the wide response of the lake boundary effectively but also distinguish the pixel values near the lake boundary from the pixel values inside the lake.

## 3. Results

We designed experiments to compare the results of the proposed method with the results of the object-oriented method. We took the study area 1 of Landsat TM image as an example to explain the process of lake extraction by the proposed method briefly: first, we segmented the band composite images by multi-resolution segmentation, the segmentation scale is 10. Then, we assigned classes on the segmented images according to the feature rule sets, the feature parameters of which are NDWI, slope and near-infrared band. After that, we corrected the lake boundary with the seed growth method. The thresholds of recognizing the lake boundary are 150 (threshold_1) and 145 (threshold_2). Figure 10 shows the results of each stage according to the above process.

Taking the visual interpretation results of lakes in the experimental area as references, we calculated the user accuracy (*UA*), producer accuracy (*PA*), overall accuracy (*OA*),

commission error (*CE*) and omission error (*OE*) of the two methods by the confusion matrix. According to the confusion matrix, the overall accuracy of classification is defined as:

$$OA = \frac{(a+d)}{(a+b+c+d)} \tag{4}$$

where *a* represents the number of lake pixels those are classified correctly, and *d* represents the number of non-water pixels those are classified correctly. Given that our experiments only extract the lakes in the study area and the other areas are all merged into a non-water area, if we directly use the mentioned definition to calculate the overall accuracy, the size of the selected study area will cause a great impact on the results. Consider an extreme case when *d* >> *a*, the overall accuracy tends to be 100%. Therefore, the overall accuracy of the result under this definition of calculation is always very high, which does not have enough reference value. Therefore, we adjusted the calculation method and defined the overall accuracy of the result as follows:

$$OA_1 = \frac{a}{(a+b+c)} \tag{5}$$

where *a* represents the number of correctly classified pixels of lakes, *b* and *c* represents the number of the wrongly classified and omitted pixels of lakes.

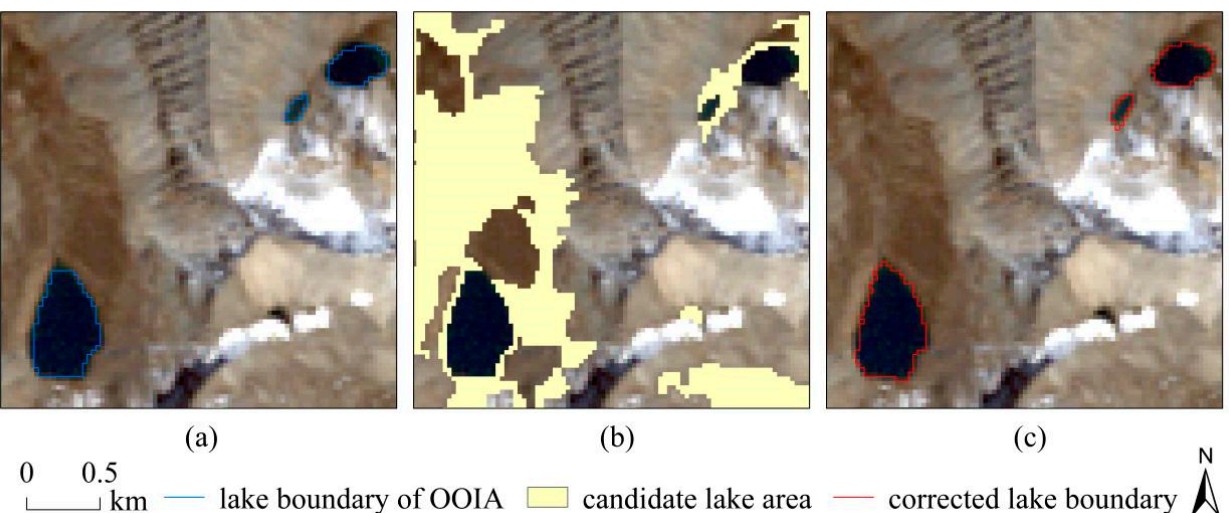

0    0.5
⌞______⌟ km    —— lake boundary of OOIA    ▭ candidate lake area    —— corrected lake boundary    N ⤒

**Figure 10.** Results of each stage of extraction process: (**a**) result of the object-oriented method; (**b**) candidate lake area obtained by the object-oriented method; (**c**) the result after the correction of the lake boundary.

According to the above extraction process, we selected the areas with densely distributed lakes in study area 1 and study area 2 as the experimental areas. The segmentation scales are all 10, except for the image of Landsat OLI, which is 75.

Figure 11 shows the examples of the results with proposed method (OOBR). The accuracy of the results is shown in Table 3.

From the results in Table 3, we can see that the overall accuracy of the results by the proposed method is more than 95%, with a maximum of 98.5%. The omission error is generally low, and the commission error is relatively higher, which may be related to the threshold selected in the process of boundary recognition; it also shows the limitations of the proposed method. However, combined with the analysis of the lake boundaries in Figure 11, it can be found that the lake boundary obtained by the proposed method is morphologically reasonable, and the commission error has no obvious impact on the morphological characteristics of the lake boundaries.

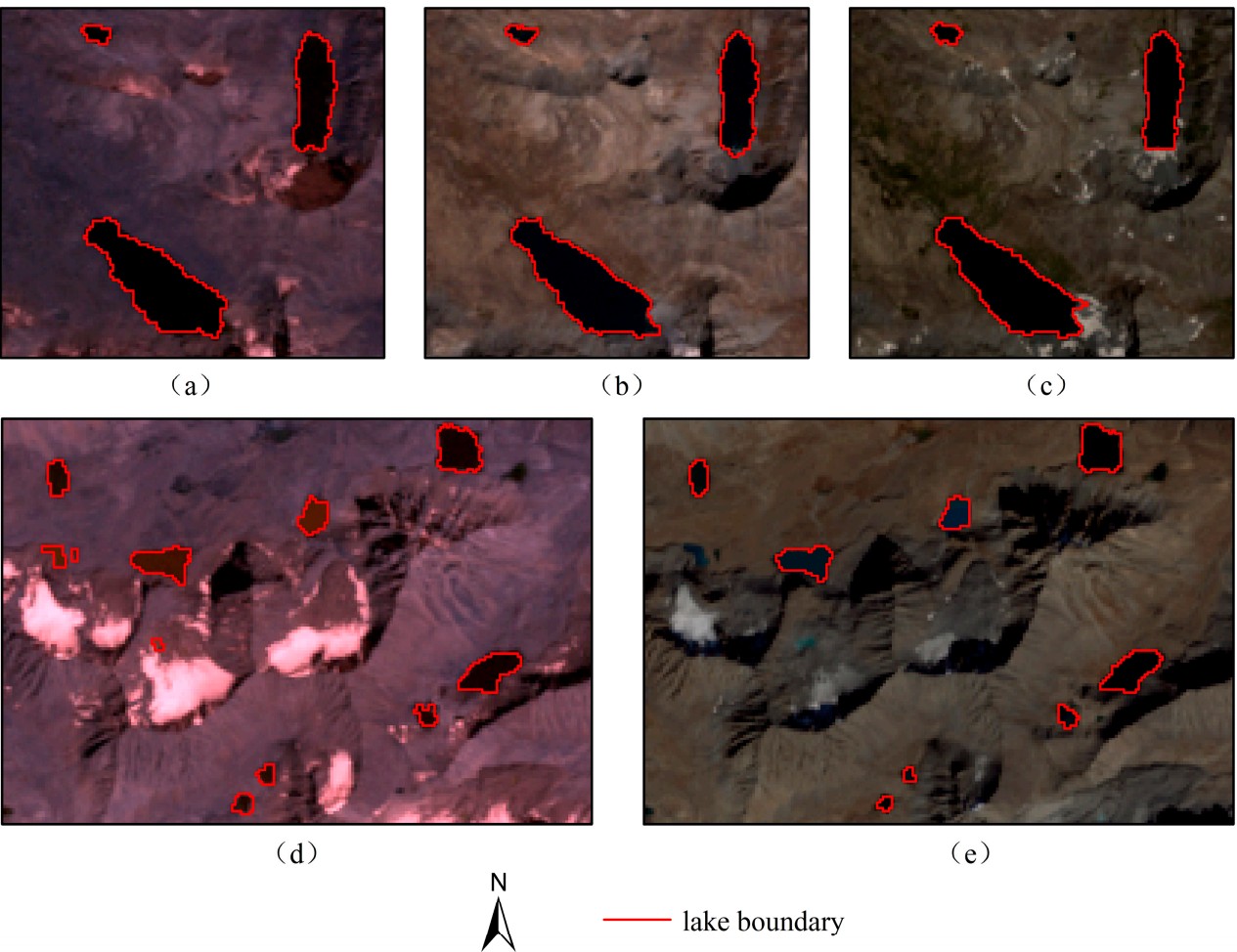

（a）　（b）　（c）

（d）　（e）

N

—— lake boundary

**Figure 11.** Extraction results with OOBR: (**a**–**c**) are the results of Landsat TM, ETM+ and OLI images, respectively, in study area 1; (**d,e**) are the results of Landsat TM and OLI images, respectively, in study area 2.

**Table 3.** Accuracy of the results with OOBR.

| Image | *UA*/% | *PA*/% | *OA$_1$*/% | *CE*/% | *OE*/% |
|---|---|---|---|---|---|
| LT5_1 | 99.00 | 99.49 | 98.50 | 1.00 | 0.51 |
| LE7_1 | 98.78 | 99.48 | 98.27 | 1.22 | 0.52 |
| LC8_1 | 99.68 | 98.47 | 98.16 | 0.32 | 1.53 |
| LT5_2 | 98.78 | 99.04 | 97.84 | 1.22 | 0.96 |
| LC8_2 | 97.95 | 98.23 | 96.25 | 2.05 | 1.77 |

## 4. Discussion

### 4.1. Comparison of Object-Oriented Method and the Proposed Method

Based on the above experiment, we designed experiments to compare the accuracy of the results of the object-oriented method and the proposed method, and verify the correction effect of the proposed method on the lake boundary. The data and parameters in this experiment are the same as those in the above experiment, and the feature rule set for the same experimental area in the two methods also have no change. Figure 12 and Table 4 shows the results.

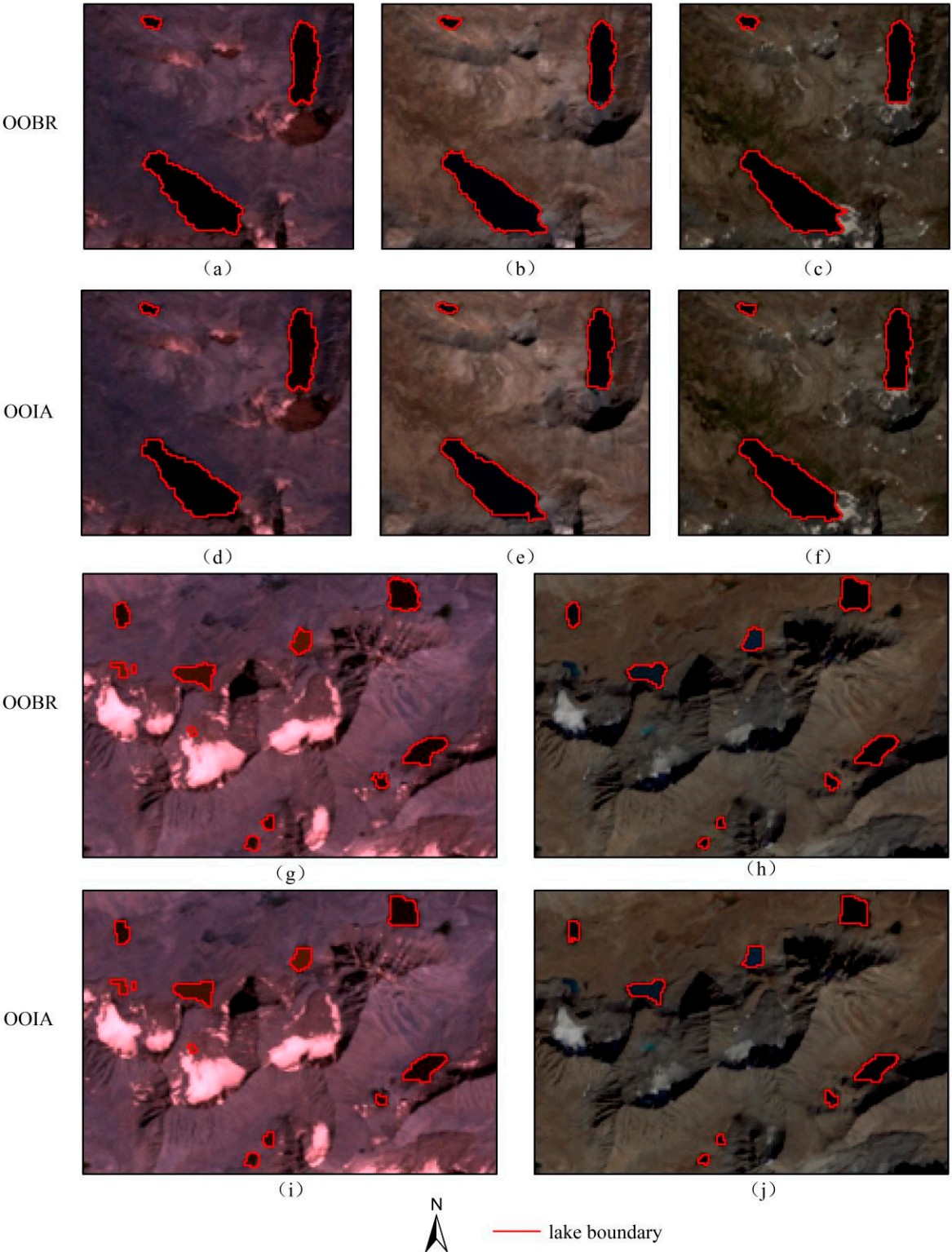

**Figure 12.** Extraction results with OOIA and OOBR: (**a**–**c**) are the results of Landsat TM, ETM+ and OLI images, respectively, in study area 1 by proposed method (OOBR); (**d**–**f**) are the results of Landsat TM, ETM+ and OLI images, respectively, in study area 1 by object-oriented method (OOIA); (**g**,**h**) are the results of Landsat TM and OLI images, respectively, in study area 2 by proposed method (OOSR); (**i**,**j**) are the results of Landsat TM and OLI images, respectively, in study area 2 by object-oriented method (OOIA).

**Table 4.** Accuracy of the results with OOIA and OOBR.

| Method | Image | UA/% | PA/% | OA$_1$/% | CE/% | OE/% |
|--------|-------|------|------|----------|------|------|
| OOBR | LT5_1 | 99.00 | 99.49 | 98.50 | 1.00 | 0.51 |
| | LE7_1 | 98.78 | 99.48 | 98.27 | 1.22 | 0.52 |
| | LC8_1 | 99.68 | 98.47 | 98.16 | 0.32 | 1.53 |
| | LT5_2 | 98.78 | 99.04 | 97.84 | 1.22 | 0.96 |
| | LC8_2 | 97.95 | 98.23 | 96.25 | 2.05 | 1.77 |
| OOIA | LT5_1 | 99.08 | 93.48 | 92.68 | 0.92 | 6.52 |
| | LE7_1 | 99.25 | 90.08 | 89.47 | 0.75 | 9.92 |
| | LC8_1 | 98.79 | 89.81 | 88.84 | 1.21 | 10.19 |
| | LT5_2 | 98.71 | 93.42 | 92.29 | 1.29 | 6.58 |
| | LC8_2 | 99.31 | 87.31 | 86.78 | 0.69 | 12.69 |

In Table 4, the highest overall accuracy of the proposed method is up to 98.5%, whereas that of the object-oriented method is 92.68%. The lowest and highest omission errors of the object-oriented method are 6.52% and 12.69%, respectively, which are both higher than the highest omission error of the proposed method.

The overall accuracy of lake extraction of the proposed method is higher than that of the object-oriented method. The results with the object-oriented method have higher omission errors, which is related to the multi-resolution segmentation in the object-oriented method: the images are segmented into "object patches" and the lake region is the merged result of the "object patches", which causes careless classification of the pixels near the lake boundary. In Figure 12, the number of omitted pixels in the results with the proposed method are obviously fewer, which is because the proposed method recognizes the lake boundary on the basis of the extraction results from the object-oriented method, which appends the omitted pixels of the lake to the lake region.

*4.2. Influence of Segmentation Scale*

The analysis of the above experiment suggests that the extraction results with high omission errors by the object-oriented method are mainly due to the multi-resolution segmentation. In order to explore the influence of different segmentation scales on the results, we selected three different segmentation scales to extract the lakes by the proposed method and the object-oriented method in the areas where the lakes are densely distributed in study area 1.

In order to follow the principle of variable-controlling, the feature rule sets of the same method with different segmentation scales are the same. The feature rule sets of the two methods are both built relatively best for the experiments with the segmentation scale of 10. We used the Landsat TM images of study area 1 in this experiment, and the segmentation scales of the three groups of experiments are 10, 12, and 15, respectively. Table 5 shows the accuracy of the experimental results.

**Table 5.** Accuracy of the two methods with different segmentation scales.

| Method | Scale | UA/% | PA/% | OA$_1$/% | CE/% | OE/% |
|--------|-------|------|------|----------|------|------|
| OOBR | 10 | 99.12 | 95.65 | 94.84 | 0.88 | 4.35 |
| | 12 | 92.33 | 98.94 | 91.42 | 7.67 | 1.06 |
| | 15 | 92.44 | 97.24 | 90.07 | 7.56 | 2.76 |
| OOIA | 10 | 98.92 | 92.36 | 91.44 | 1.08 | 7.64 |
| | 12 | 98.80 | 92.10 | 91.08 | 1.20 | 7.90 |
| | 15 | 97.03 | 91.88 | 89.37 | 2.97 | 8.12 |

According to the results, the overall accuracy of the proposed method is always higher than that of the object-oriented method with different segmentation scales. The overall accuracy of the proposed method with the segmentation scale of 10 is up to 94.84% and

that of the object-oriented method is only 91.44%. When the overall accuracy of the object-oriented method is 89.37% with the segmentation scale of 15, the overall accuracy of the proposed method is 90.07%, which is still higher than that of the object-oriented method.

The data in Table 5 shows that the overall accuracy of the two methods both decrease with the increase in the segmentation scale. It can be inferred from the commission error and the omission error that the decrease in the overall accuracy of the two methods is mainly due to the increase in the commission error.

In the result, we found that the commission error increased strongly when the segmentation scale changes from 10 to 12, whereas as the segmentation scale continuously increases from 12 to 15, the commission error is relatively stable. The process of lake boundary correction proposed in this paper could explain this phenomenon: the seeds from the object-oriented method may expand with the increase in the segmentation scale, which may cause that the boundary of seed is at the position of type *c* or *d*, where the pixel meeting the thresholds cannot be found with the process of seed growth. In this case, the process of seed growth will not stop until the pixel value is 0 in the merged image. However, fortunately, it is this constraint of seed growth that limits the commission error to a relatively low level with the increase in the segmentation scale. As for the results of the object-oriented method, with the increase in the segmentation scale, not only the omission errors still remain at a high level, but also the commission error of the results increases; thus, the overall accuracy of the results is still lower than that of the proposed method.

Figure 13 shows the comparison of the results of two methods with different segmentation scales:

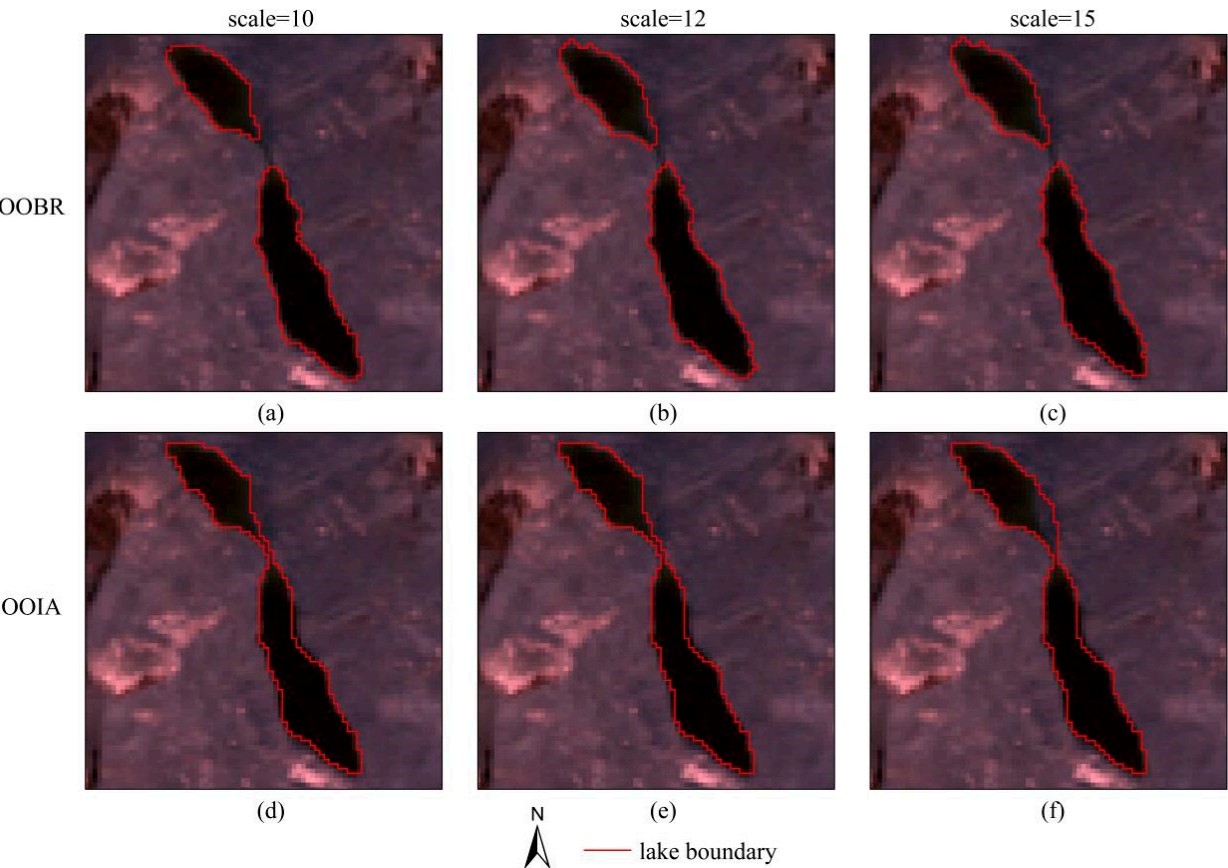

**Figure 13.** Results of the two methods with different segmentation scales image: (**a**–**c**) are the results using OOBR method with segmentation scale settings of 10, 12 and 15, respectively; (**d**–**f**) are the results using OOIA method with segmentation scale settings of 10, 12 and 15, respectively.

When the segmentation scale is 10, there are many obvious omitted pixels of lake in the results using the object-oriented method. With the increase in the segmentation scale, there are many misjudged pixels of lake in the results. Therefore, it is difficult to obtain a more accurate lake extraction result with the change of the segmentation scale by the object-oriented method. With the change of segmentation scale, the results of the proposed method are relatively stable. Although the commission error of the results increases to a certain extent, the overall accuracy is higher than 90% and the morphology of the extracted lake is reasonable.

The morphology of the extracted lake by the proposed method is more robust to the change of the segmentation scale than that of the object-oriented method. Additionally, the overall accuracy of the proposed method has not been lower than that of the object-oriented method with the increase in the segmentation scale.

### 4.3. Comparison of Results with the Watershed Algorithm and the Proposed Method of Correction

In order to compare the difference of the results from the proposed method of correction and the watershed algorithm, we also designed the following experiment: we used the proposed method (OOBR) and the method of combining object-oriented and watershed algorithm (OOW) to extract lakes, respectively in the same region in study area 1 of the Landsat TM image. The segmentation scale is 12; Figure 14a,b shows the two feature rule sets; Figure 15 shows the results.

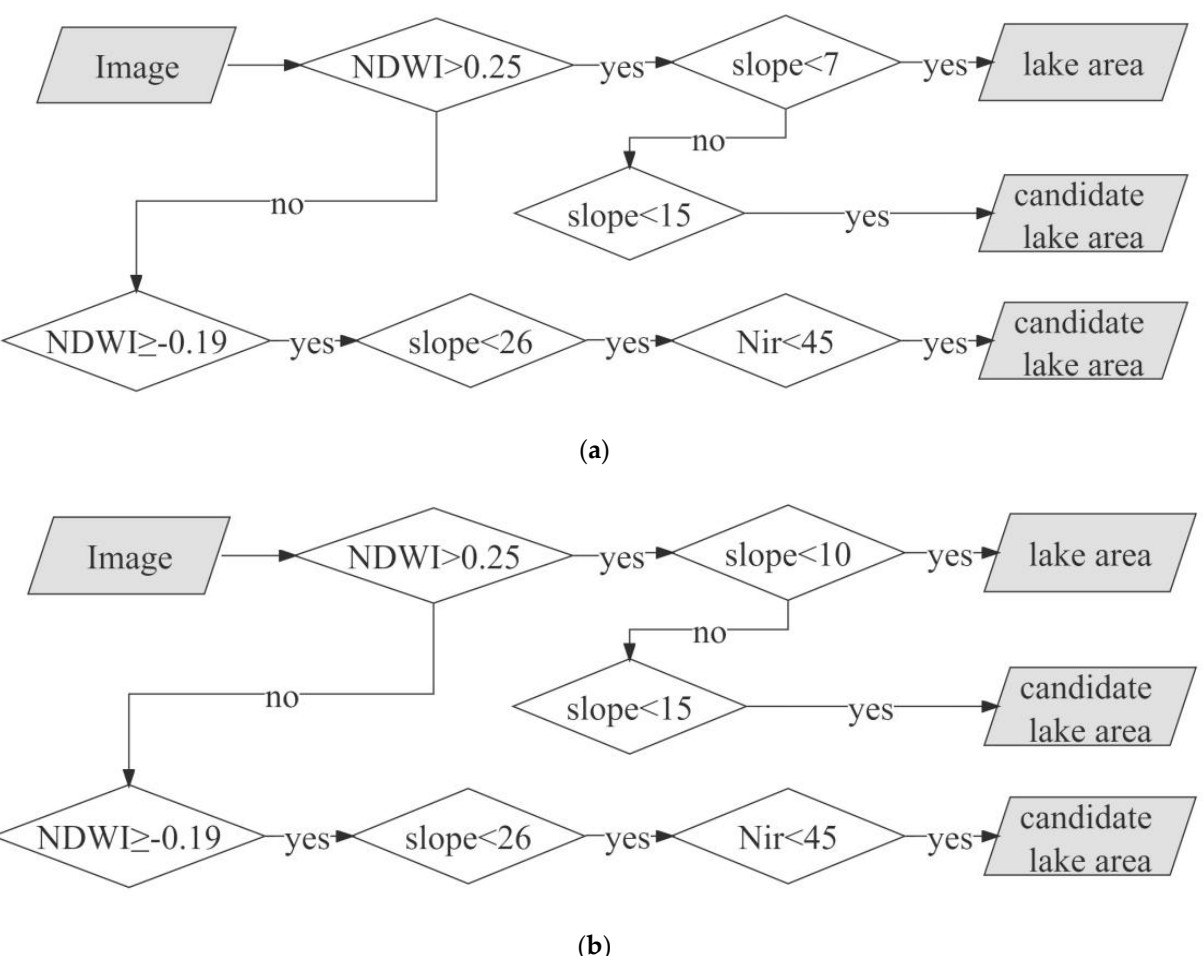

**Figure 14.** Feature rule sets: (**a**) one feature rule set in the comparison experiment; (**b**) another feature rule set in the comparison experiment.

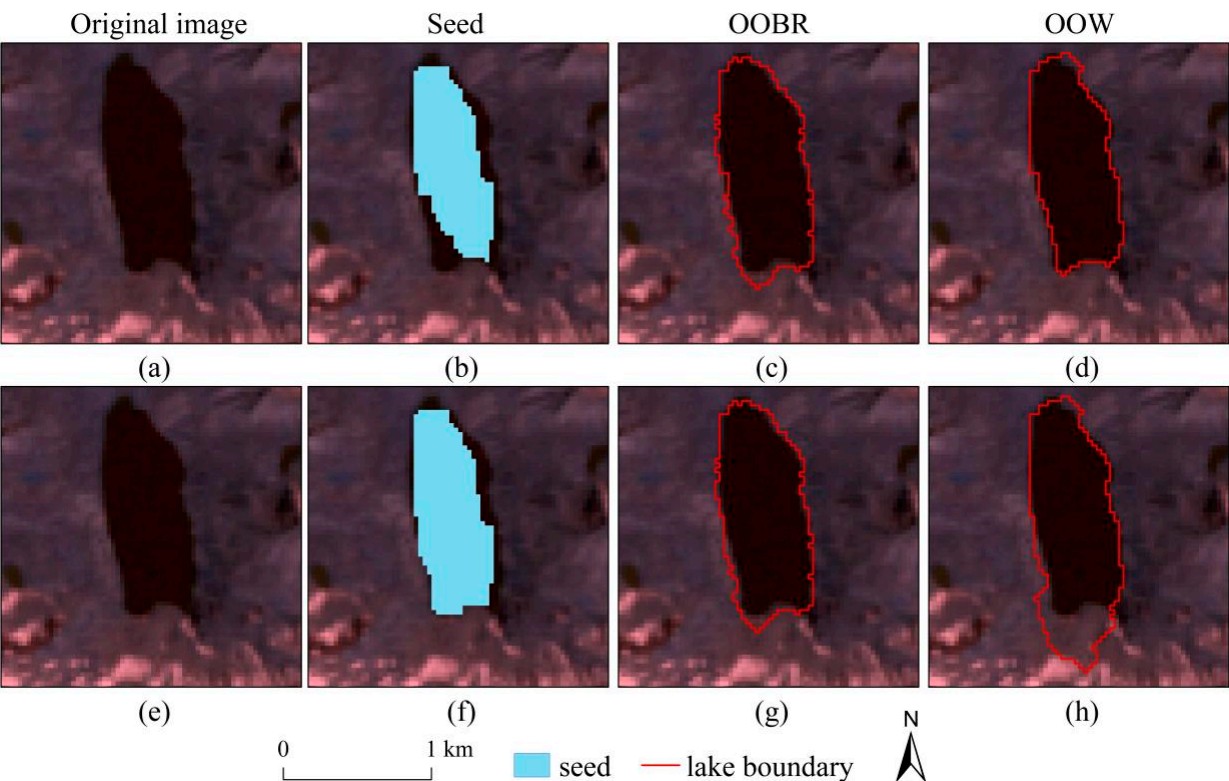

**Figure 15.** Results of the method of OOBR and OOW with different feature rule sets image: (**a**,**e**) are the original remote sensing images from the experimental area; (**b**–**d**) are the results with the feature rule set in Figure 14a; (**f**–**h**) are the results with the feature rule set in Figure 14b.

In Figure 15, with different seeds from the object-oriented stage, the results of OOBR are stable, but the results of OOW are opposite. The result of feature rule set (a) of OOW is acceptable, but there is an obvious commission error with the feature rule set (b). It can be seen that the two seeds are both reasonable, if the seed is larger, the recognition of the lake boundary by the watershed algorithm will become terrible. The method of OOBR is more robust to the seed.

The reason for the big commission error of OOW is that the coverage of the seed exceeds the range of the highlighted lake area in the MNDWI image in local regions, which cannot be completely avoided by this method. The seed is unstable with the change of segmentation scale and feature rule set. It is hard to guarantee that the seeds are all completely contained in the highlighted regions, regardless the highlighted area is in the image of MNDWI or NDWI or some other water indexes. It is important to find a method of lake boundary recognition which is robust to the unstable seed obtained in the object-oriented stage. In the proposed method of OOBR, we limited the process of lake boundary recognition to the symmetrical area in the merged image, which would not lead to a large commission error no matter how the seed changes, as long as it is reasonable.

### 4.4. Verification of the Universality of Proposed Method

To illustrate the universality of the proposed method, we chose the study area 4 in the China–Pakistan Economic Corridor to carry out the lake extraction experiment. Among the three remote sensing images in this region, the segmentation scales of TM and ETM images are 10, and that of OLI image is 75. The examples of experimental results are shown in Figure 16.

Landsat TM          Landsat ETM+          Landsat OLI

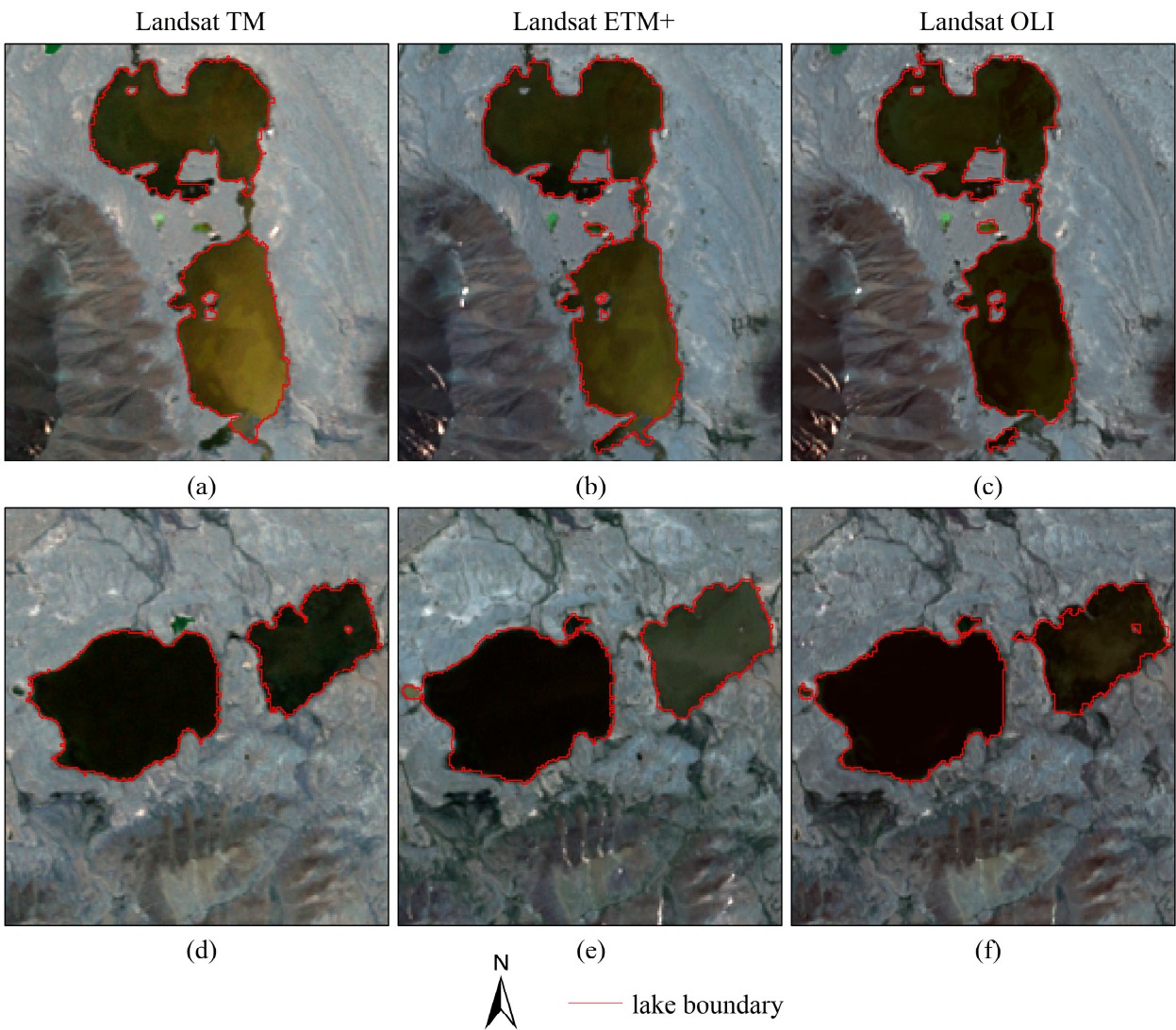

**Figure 16.** Results of the proposed method in study area 4: (**a**,**d**) are the results in the image of Landsat TM; (**b**,**e**) are the results in the image of Landsat ETM+; (**c**,**f**) are the results in the image of Landsat OLI.

Table 6 presented the accuracy of the results in study area 4. We can see that the overall accuracy reached more than 95%, the commission errors are less than 2.5%, and the omission errors are less than 1.5%. Figure 16 showed that the method has a good response to the impact of the island in the center of the lake. Some of the small lakes have not been identified in this study area, which is related to the extraction of seeds in the object-oriented method at the first step. By adjusting the parameters of the object-oriented method, the accuracy of lake extraction can be further improved.

**Table 6.** Accuracy of the results in study area 4 using the proposed method.

| Image | UA/% | PA/% | $OA_1$/% | CE/% | OE/% |
|---|---|---|---|---|---|
| Landsat TM | 97.52 | 99.46 | 97.01 | 2.48 | 0.54 |
| Landsat ETM+ | 99.73 | 99.33 | 99.06 | 0.27 | 0.67 |
| Landsat OLI | 98.21 | 98.66 | 96.92 | 1.79 | 1.34 |

## 5. Conclusions

In order to monitor the glacier lakes in the China–Pakistan Economic Corridor region, where the lake expansion is accelerated with global warming, in this paper we proposed a lake extraction method which combines the object-oriented method with boundary recognition (OOBR). Based on the result (seed) obtained by the object-oriented method, we introduced the thresholds of recognition in the symmetrical area to correct the lake boundary, and the lake extraction accuracy are improved. Finally, we can obtain the following conclusions:

(1) The proposed method can reduce the omission error of the results by the object-oriented method, and the highest overall accuracy of the proposed method is 98.5%.
(2) The overall accuracy of the proposed method is always higher than that of the object-oriented method with a reasonable change of segmentation scale, and its overall accuracy remains above 90% in the experiments.
(3) Compared with the method combining the object-oriented method with the water-shed algorithm, the proposed method is more robust to the preliminary extraction result (seed).

In this paper, the introduction of symmetry of lake boundary makes up for the defects of object-oriented method brought by multi-resolution segmentation in some extent, and it effectively limits the accuracy of lake extraction to a higher level, which makes the extraction result reasonable and accurate.

Additionally, we can further improve the method from the following aspects:

(1) The time complexity of the lake boundary recognition in the symmetrical area can be further optimized.
(2) In the method of determining the weights, we assumed that the two weights are both integers and their sum is 10. We selected the relatively optimal weights through "trail and test" experiments, so the results may not be as good as the analytical solution obtained by some optimization algorithms.
(3) The generalization of the thresholds of the lake boundary recognition is limited, which may be optimized by considering the adaptive threshold method.
(4) The process of lake boundary correction may cause the commission error in some extent, which can be improved by considering the topological relationship between the pixels that satisfy the thresholds of recognition and the preliminary extraction.

In our future research work, we will consider the factors we mentioned above to further improve the accuracy of lake extraction.

**Author Contributions:** Conceptualization, W.W.; methodology, W.W., W.L. and B.L.; validation, B.L.; formal analysis, B.L.; writing—original draft preparation, B.L.; writing—review and editing, W.L. and B.L.; visualization, B.L.; supervision, W.W. and W.L.; project administration, W.W.; funding acquisition, W.W. All authors have read and agreed to the published version of the manuscript.

**Funding:** This research was funded by Key Project of Innovation LREIS [grant number: KPI007] and National Nature Science Foundation of China, [41421001].

**Data Availability Statement:** The data presented in this study are available on request from the corresponding author. The data are not publicly available due to the privacy restrictions.

**Conflicts of Interest:** The authors declare no conflict of interest.

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
