# Peer review of "A Lake Extraction Method Combining the Object-Oriented Method with Boundary Recognition"

_land, doi:10.3390/land12030545_

Round 1

Reviewer 1 Report

I would like to thanks for inviting me to review this manuscript entitled "A lake extraction method combining the object-oriented method with boundary recognition ". 

The use of Object oriented image analysis with boundary recognition  is particularly noteworthy, as it provides a robust and thorough analysis of the subject. The results are clearly presented and support the conclusions that are drawn.

"I have thoroughly reviewed the methodology used in this paper, and I have a few suggestions for improvement.

First, I would suggest the feature extraction and post classification for the same. This could provide a more robust analysis and help to address the subject. 

Additionally, I noticed that the methodology does not take into account how the weights have given for particular area. This could be addressed by incorporating particular methodology like Analytical heirarchial process or multi criteria decision making (MCDM).

 By incorporating the suggestions mentioned above, I believe that the results could be further strengthened and the conclusions more robust.

Author Response

Dear Reviewer,

Thank you for your letter and for the comments concerning our manuscript entitled “A lake extraction method combining the object-oriented method with boundary recognition” (manuscript ID: land-2198189). We greatly appreciate your revision and advice on our manuscript. Those comments are all valuable and very helpful for revising and improving our paper, as well as the important guiding significance to our researches. The suggestions/recommendations have been carefully considered, we have made the necessary changes related to these comments in the enclosed revised manuscript. Revised portion are marked in red in the file named “Manuscript-Track Changes”. The responses to your comments are in the attached file named “Response to reviewer 1 comments”.

Kind regards,

Sincerely yours,

Bingxue Liu

Wei Wang

Institute of Geographic Sciences and Natural Resources Research, CAS

11A, Datun Road, Chaoyang District, Beijing, 100101, China

Tel:+86-135-2011-2446

E-mail: wang_wei@lreis.ac.cn

Reviewer 2 Report

The article describes the method of the automatic lake extraction. The authors combined the object-oriented image analysis with boundary recognition and tested it in the  The China-Pakistan Economic Corridor (3 study areas). The method is well described and analyzed.

I have some comments:

1. Line 9. “the” is doubled.

2. Figure 1. I prefer to add a more global map, as it is unclear where the area is situated.

3.  Table 1. It is better to sort the images by date.

4. Figure 3. It is necessary a more detailed figure caption.

5. Figure 4. It is necessary to add labels for axis.

6. Line 186. The dimensions of thresholds are unclear.

7. Figure 11. It is necessary a more detailed figure caption.

8. In Discussion I would like to see a comparison with another methods of lake extraction and an application for another regions (or requirements for landscape where proposed method works well).

Author Response

Dear Reviewer,

Thank you for your letter and for the comments concerning our manuscript entitled “A lake extraction method combining the object-oriented method with boundary recognition” (manuscript ID: land-2198189). We greatly appreciate your revision and advice on our manuscript. Those comments are all valuable and very helpful for revising and improving our paper, as well as the important guiding significance to our researches. The suggestions/recommendations have been carefully considered, we have made the necessary changes related to these comments in the enclosed revised manuscript. Revised portion are marked in red in the file named “Manuscript-Track Changes”. The responses to your comments are in the attached file named “Response to reviewer 2 comments”.

Kind regards,

Sincerely yours,

Bingxue Liu

Wei Wang

Institute of Geographic Sciences and Natural Resources Research, CAS

11A, Datun Road, Chaoyang District, Beijing, 100101, China

Tel:+86-135-2011-2446

E-mail: wang_wei@lreis.ac.cn

Reviewer 3 Report

The paper is well written and organized and deals with an interesting topic. The results highlight the performance of the proposed methodology. Some minor revisions:

-      “Recent years, the object-oriented extraction method has gradually replaced the pixel- level extraction method in the lake extraction with remote sensing data “ à Please provide a reference.

- Please provide more references of other methods that apply modern machine learning / deep learning techniques . Please provide a discussion on this as well as the differences /approach/advantages-disadvantages with the object based methods (and to the proposed method) 

-      Please highlight the innovation features of the proposed methodology in a sub-section.

Author Response

Dear Reviewer,

Thank you for your letter and for the comments concerning our manuscript entitled “A lake extraction method combining the object-oriented method with boundary recognition” (manuscript ID: land-2198189). We greatly appreciate your revision and advice on our manuscript. Those comments are all valuable and very helpful for revising and improving our paper, as well as the important guiding significance to our researches. The suggestions/recommendations have been carefully considered, we have made the necessary changes related to these comments in the enclosed revised manuscript. Revised portion are marked in red in the file named “Manuscript-Track Changes”. The responses to your comments are in the attached file named “Response to reviewer 3 comments”.

Kind regards,

Sincerely yours,

Bingxue Liu

Wei Wang

Institute of Geographic Sciences and Natural Resources Research, CAS

11A, Datun Road, Chaoyang District, Beijing, 100101, China

Tel:+86-135-2011-2446

E-mail: wang_wei@lreis.ac.cn
